# Development and Reproduction of a Japanese Strain of *Ctenolepisma calvum* (Ritter, 1910) at Room Temperature

**DOI:** 10.3390/insects14060563

**Published:** 2023-06-16

**Authors:** Hiroki Watanabe, Megumi Shimada, Yoshinori Sato, Rika Kigawa

**Affiliations:** 1Kyushu National Museum, Dazaifu 818-0118, Japan; kigawa-r@kyuhaku.jp; 2Tokyo National Research Institute for Cultural Properties, Tokyo 110-0007, Japan; shimada-m4a@nich.go.jp (M.S.); sato-y8s@nich.go.jp (Y.S.)

**Keywords:** *Ctenolepisma calvum*, Lepismatidae, insect development, head width, reproduction, museum pests, household pests

## Abstract

**Simple Summary:**

*Ctenolepisma calvum*, commonly known as the ghost silverfish, is an insect that lives in buildings, including museums, libraries, and archives. It is regarded as a pest of paper-based objects. Recently, *C. calvum* was discovered in several areas of Japan for the first time, and it may be a new threat to the conservation of collections that are of cultural and historical importance. Because the biological characteristics of this species in Japan are not well known, we observed their growth and reproduction. We found that eggs were laid from April to November, especially in early June, and eggs usually hatched within two months. The young insects grew through molting, and started laying eggs the next year. Females laid around 10 or more eggs at one time, and they were able to lay eggs once or more per year. Through this study, only females were found, and they were able to reproduce without male individuals. Further research on practical control methods for this species in museums and other facilities is required.

**Abstract:**

*Ctenolepisma calvum* (Ritter, 1910) (Zygentoma: Lepismatidae) is a primitive wingless insect that causes damage to paper, and it is regarded as a pest of collections in museums, archives, and libraries. This species was recently discovered in Japan for the first time and may have already spread over large areas of Japan, but, currently, no information is available on the biological characteristics of *C. calvum* in Japan. In this study, we observed the processes of development and reproduction of *C. calvum* found in Japan at room temperature. Oviposition was observed from April to November, with a peak in early June. The average egg period was 56.9 days at average temperatures above 24.0 °C, and was 72.4 days at average temperatures below 24.0 °C. The 1st, 2nd, and 3rd instars lasted 4.7 days, 13.2 days, and 26.1 days on average, respectively, at average temperatures above 22.0 °C. Average instar periods were 23–28 days in 4th–7th instars and tended to increase in later instars. Instar periods also increased when the average temperature was 22.0 °C or lower. In individual rearing, the longest-living individual lived for approximately two years, up to the 15th instar. The head width grew at an approximate ratio of 1.1 per molt. First oviposition occurred at the 10th or 11th instar. Individually observed females oviposited once or twice a year, laying 6–16 eggs at one time, but females at least two years old laid 78.2 eggs per year on average in a mass-culture cage. Through this study, only females were found, and the mature females produced their progenies parthenogenetically.

## 1. Introduction

Zygentoma are an order of primitive wingless insects that include species commonly known as silverfish and firebrats. Zygentoma are ametabolous insects, meaning that juveniles and adults are morphologically similar and adults continue molting after they reach sexual maturity [1]. Some species of Zygentoma live indoors and are infamous for feeding on paper; they can cause irreversible damage to books, scrolls, and other paper-based objects that are of cultural and historical importance. Therefore, it is essential for museums, archives, and libraries to implement measures for preventive conservation.

In Japan, species of Zygentoma that are known as pests of cultural properties include *Ctenolepisma villosum* (Fabricius, 1775), *Lepisma saccharinum* Linnaeus, 1758, *Thermobia domestica* (Packard, 1873), *C. lineatum* (Fabricius, 1775), and *C. longicaudatum* Escherich, 1905, all of which belong to the family Lepismatidae [2,3]. Recently, another species, *C. calvum* (Ritter, 1910), was discovered in several areas of Japan for the first time [4].

*Ctenolepisma calvum* was first described in Ceylon (now Sri Lanka) by Ritter [5] in 1910, and was also found in Central America (Guyana and Cuba) [6]. Recently, it has spread rapidly in European countries, such as Austria, Czechia, Germany, and Norway [7,8,9,10]. Shimada et al. [4] summarized that, as of 2022, it has been recorded in at least 25 countries around the world (including some records online).

According to Shimada et al. [4], *C. calvum* has already been found in several museums and libraries in Japan. They also pointed out that no males were found among the captured individuals and bred populations, and that this species might reproduce parthenogenetically [4]. If this is the case, *C. calvum* should have high fecundity, meaning that their population can grow rapidly, presenting a new threat to the conservation of cultural properties in Japan. However, currently, information on the biological characteristics of *C. calvum*, which is important for implementing effective pest management in museums and other institutions, is scarce, and no data are available on the life history of *C. calvum* in Japan. In this study, we observed the processes of development and reproduction of a strain of *C. calvum* found in Japan at room temperature and obtained fundamental data, including the development period of eggs and each instar, survival rates, changes in body size, and fecundity.

## 2. Materials and Methods

In this study, a laboratory strain of *C. calvum*, which originated from individuals collected in Fukuoka Prefecture, Japan, was used for the experiments. The strain was mass-cultured in plastic cages containing facial tissue (e.g., Elleair, Daio Paper Corporation, Tokyo, Japan), powdered dry crickets (field-collected), and soybean flour as diets.

For the observation of eggs and early instar juveniles, a total of 70 eggs of known deposition dates were collected from mass-culture cages. The eggs were normally found between the layers of facial tissue or at the bottom of the cages. Using a soft writing brush, the eggs were placed in petri dishes (diameter 64 mm and depth 21 mm [including lid]) containing a sheet of facial tissue. Approximately 10 eggs were placed in one petri dish. Powdered dry crickets, soybean flour, and fecal pellets collected from mass-culture cages were added in the petri dishes as diets. Fecal pellets were included because some studies suggested that in *T. domestica*, a Lepismatid species, microbial symbionts are spread through feces [11,12]. When hatching or ecdysis events were observed, the individuals were photographed using a digital microscope (VMM-100, Visualix, Kobe, Japan), and the head width (HW) between their compound eyes was measured. Normally, such analysis is based on the head capsule width, but, in this study, we used the distance between the compound eyes, following a precedent study on the development of a *Ctenolepisma* sp. (specific name not given) [3]. Specifically, we measured the distance between the outer edges of the compound eyes of *C. calvum* (Figure 1). The hatching rate of eggs and survival rate of early (1st–2nd) instar juveniles were calculated. Because it was difficult to determine the survival rates of 1st and 2nd instar juveniles separately, they were grouped as the survival rate of “1st–2nd instar juveniles”, i.e., the proportion of individuals that completed the 1st and 2nd instars to all newly hatched 1st instar juveniles.

The observation of development of middle to later instars was started mainly from the 3rd instar because handling juveniles at the 1st or 2nd instar seemed to injure them, often causing them to weaken and die afterward. As shown later, the 1st, 2nd, and 3rd instar juveniles were visually distinguishable. Twenty-five 3rd instar juveniles, 13 of which were newly collected from the cages and the remaining 12 individuals observed from hatching, were placed in petri dishes (which were the same type as in the observation of eggs) containing facial tissue, powdered dry crickets, soybean flour, and fecal pellets (Figure 2). One or two individuals were placed in one petri dish. Each of the individuals was observed 5–6 days a week, and when an ecdysis event was observed, the individual was photographed using the digital microscope to measure its HW. The body length of the individuals was measured once or up to three times in an instar. Individual identification in a petri dish containing two individuals was possible based on the time of ecdysis and HW measurements, but later they were individually placed in separate petri dishes before the 10th instar at latest. After the 7th instar, each individual was also observed from underneath the petri dish, using a portable stereomicroscope (Fieldmicroscope, Nikon Corporation, Tokyo, Japan), to confirm the presence of an ovipositor. Instar-specific survival rates were calculated for 3rd and later instars. To supplement data on juvenile development from the 1st to 3rd instars, nine 1st instar juveniles and six 2nd instar juveniles were additionally observed until the 2nd–4th instars.

When oviposition occurred in the petri dishes for the observation of development, the eggs were counted and collected. To further obtain information regarding the seasonal trend of oviposition, eggs laid in mass-culture cages were also collected and recorded throughout 2020. In early April 2020, the cages and petri dishes contained approximately 80–90 individuals in total, but it was unknown whether all of them were at the reproductive stage or not or when the individuals died (the number of live individuals decreased with time). For this reason, observation of oviposition trend was conducted again in 2022. We prepared two cages, one containing five females born in 2020 or before, and the other containing 12 females born in 2021. All 17 females were estimated to be sexually mature, and all lived through 2022, except for one female, which was born in 2021 and died in July 2022. The HW ranges of the females born in 2020 or before and in 2021 were 1.18–1.24 and 1.01–1.09, respectively, as of December 2022.

All mass-culture cages and petri dishes for the observation were kept in a larger clear polypropylene box containing a bag of moisture stabilizer (Artsorb^®^, Fuji Silysia Chemical Ltd., Kasugai, Japan) in a laboratory room. The temperature was unspecified and was allowed to follow that in the room. As a result, the temperature was kept mostly at 23–26 °C, but it dropped to around 20 °C in the winter season (Figure 3). The relative humidity (RH) was maintained mostly at 60–70% (Figure 3). Light conditions, similarly, were not strictly controlled, but the room was usually lit from 07:00 a.m. to 21:00 p.m., during which time the illuminance level in the polypropylene box was 40–90 lux.

## 3. Results

### 3.1. Developmental Period and Survival Rate of Eggs

The eggs (Figure 4a) were elliptic, 0.90 ± 0.02 mm (mean ± se) long, and 0.65 ± 0.02 mm wide (*n* = 10). The eggs hatched normally within two months, but the egg periods were longer when the room temperature dropped in winter. Table 1 shows the average egg periods and survival rates of eggs and 1st–2nd instar juveniles (refer to Appendix A for the original data). For a simple calculation of the average egg period, we set a single border of temperature at 24.0 °C, below which the egg periods tended to lengthen. The average egg period was 56.9 ± 0.3 days when the average temperature was above 24.0 °C, and it was 72.4 ± 1.5 days when the average temperature was below 24.0 °C. The hatching rate (survival rate of eggs) was 96%, and the survival rate of 1st–2nd instar juveniles (the proportion of individuals that completed the 1st and 2nd instars to all newly hatched 1st instar juveniles) was 78%. The survival rates were calculated without separation by temperature.

### 3.2. Development from the 1st Instar

Table 2 shows the period of each instar of each individual (refer to Appendix A for the complete set of data containing the time of ecdysis events and average temperature during each instar). Average periods of each instar are shown in Table 3. Because the instar periods tended to lengthen largely when the temperature dropped to 22.0 °C or lower in winter, average periods of each instar are presented for both periods, in which the average temperature was above 22.0 °C and was 22.0 °C or lower. Table 3 also shows the average HW for each instar and the growth ratios of HW of successive instars, the approximate range of body size at each instar, and the instar-specific survival rates at 3rd and later instars, calculated based on Table 2. It should be noted that the sample size was not necessarily equal for each row in Table 3 (refer to the Appendix A for details).

The 1st instar juveniles (Figure 4b) were not actively mobile and were not observed to feed. They molted in 4.7 ± 0.1 days at average temperatures above 22.0 °C. Their body length was approximately 1.7–2.0 mm, HW 0.360 ± 0.002 mm, and they did not have scales on the body. The 2nd instar juveniles (Figure 4c) were slightly more mobile than the 1st instar. They started feeding, and, because they had a translucent, scaleless body, the contents of the digestive system were visible. The 2nd instar lasted 13.2 ± 0.4 days at average temperatures above 22.0 °C. Their body length was approximately 1.8–2.3 mm, HW 0.398 ± 0.002 mm, and their antennae and caudal appendages became longer in proportion to their body. The 3rd instar juveniles (Figure 4d) also lacked scales, but they had longer antennae and caudal appendages in proportion to their body and were more active than the 1st and 2nd instar juveniles. Their body length was approximately 1.9–3.0 mm, HW 0.442 ± 0.002 mm. The 3rd instar lasted 26.1 ± 0.9 days when the average temperature was above 22.0 °C, and lasted 44 ± 0 days when the average temperature was 22.0 °C or lower.

The 4th instar juveniles (Figure 4e) had scales for the first time. From the 4th to 7th instars, average instar periods were 23–28 days at average temperatures above 22.0 °C. Average instar periods were 31–43 days from the 8th to 11th instars at average temperatures above 22.0 °C, and became longer afterwards. The body length and HW grew with instars, with the growth ratios of HW being around 1.1, but the growth slowed down, especially after the 12th instar (Table 3).

The largest instar number observed was 15, which was reached by individual #3. Individual #3 was estimated to have lived for approximately 23 months (exact lifespan was unknown because observation was started at the 3rd instar). Its HW was 1.155 mm at the 15th instar, and its longest body length measured was 8.59 mm, recorded at the 13th instar. In addition, the largest individual found in mass-culture had a HW and body length of 1.254 mm and 9.62 mm, respectively.

The survival rates of the 1st and 2nd instars, together, and 3rd instar were lower than those of the eggs and 4th–7th instars (Table 1 and Table 3). The survival rates also declined in the 8th–10th instars. After the 11th instar, the sample size became too small to discuss survivorship. It should be noted that the survival rates in individual rearing may be lower than in actual indoor conditions because the insects were subjected to constant observation that may have caused stress. A graph showing the trend in survivorship is shown in Appendix A.

### 3.3. Oviposition

The ovipositor first appeared at either the 8th or 9th instar, as indicated by the asterisk (*) in Table 2. At this point, ovipositors were short (Figure 5), and the individuals were probably not sexually mature. First oviposition was confirmed at the 10th or 11th instar, as indicated by two asterisks (**) in Table 2 (*n* = 5). The records of oviposition of individually reared *C. calvum* were summarized in Table 4. In an oviposition event, they laid 6–16 eggs in lots within 1–5 days. Individuals #2 and #3 laid eggs twice a year, with the second oviposition occurring 1.2–1.4 months after the first. Although there were multiple individuals that died before the sex could be identified, all individuals that reached the 9th instar were females (*n* = 14) (Table 2).

The five females in Table 4 produced eggs parthenogenetically in separate petri dishes. In detail, individuals #1 and #2 were kept in the same petri dish until the 9th instar, and they were placed in separate petri dishes, starting from the 10th instar. They both laid eggs first at the 11th instar, and individual #2 laid eggs again at the 12th instar. The same was true for individuals #3 and #4, with individual #3 undergoing the second oviposition at the 12th instar. Individual #7, which laid eggs first at the 10th instar, had been kept singly from the 3rd instar. The hatching rate of randomly chosen eggs laid by these females was 96% (*n* = 23), the same as that of the eggs collected from the mass-culture cages in Table 1. In addition, no males were found in the laboratory strain, originally collected from Fukuoka Prefecture, Japan, used in this study, nor in other strains originally collected from Tokyo Metropolis, Japan, bred by us.

Figure 6 shows seasonal changes in the number of eggs laid at 10- or 11-day intervals; the number of eggs collected from both mass-culture cages and petri dishes for individual rearing in 2020 is shown in Figure 6a, and the number of eggs laid in 2022 in two cages containing a total of 17 presumably mature females is shown in Figure 6b. Oviposition occurred from April to November, mainly from April to August, and the largest number of eggs was laid in early June. No eggs were laid from January to March or in December. In 2022, a total of 391 eggs were laid by the five females born in 2020 or before, and 175 eggs by the 12 (later 11) females born in 2021; on average, 78.2 eggs were laid per female in the former case, and 14.6 eggs per female (including one individual that died in July 2022) in the latter.

## 4. Discussion

### 4.1. Development from Egg to Adult

Based on the results, the development after hatching can be divided into at least three phases: 1st–3rd instars (growth without scale), 4th–7th instars (growth), and 8th–11th instars (growth and maturation). The developmental period of the first phase was short, with the 1st instar being the shortest of all instars, and the 2nd instar being the second-shortest (Table 3). Feeding was observed after molting to the 2nd instar. Juveniles in this phase did not have scales. They underwent a large decline in survival rates (Table 1 and Table 3, Appendix A). In the second phase (4th–7th instars), average instar periods were 23–28 days at average temperatures above 22.0 °C. The individuals had scales on the body but no ovipositors, and the survival rates were rather high. The third phase showed slower development, with average instar periods 31–43 days at average temperatures above 22.0 °C. Maturation occurred in this phase, and the survival rates declined largely again.

The first appearance of scales at the 4th instar is common to other major species of Lepismatidae, including *L. saccharinum* [13], *T. domestica* [14], and *C. longicaudatum* [15]. At constant temperatures, the instar periods tended to increase with instar numbers in these species [13,14,15], a trend that seemed to be present in *C. calvum*. In addition, in *L. saccharinum*, periods of 5th and later instars showed a very large variation at 25 °C, with maximum durations of these instars ranging from 66 to 182 days, owing to the occurrence of delayed molting [13]. In *T. domestica*, average instar periods after the 5th instar were 8 to 13 days at 37 °C [14], which are shorter and more stable than the other species.

In this study, the sample size became too small to see general patterns of development after the 11th–12th instar, but it can be inferred that instar periods increase further, and growth ratios decrease (tend to 1) to allocate resources to reproduction. In individual rearing, individual #3 lived the longest, approximately 23 months, reaching the 15th instar. Because individuals with larger HW and body length were found in mass-culture, it is possible that *C. calvum* can have more instars and live longer than two years. In addition, Shimada et al. [4] reported that breeding populations of *C. calvum* grow up to 12 mm. Reports on other species state that individuals of *L. saccharinum* can survive more than 2.5 years and molt more than 30 times [13] and *T. domestica* can have more than 42 instars and live over three years [14]. Further research is necessary on the development, lifespan, and survivor rates of *C. calvum* at the mature stages. There seemed to be no seasonal tendency in the time of death so far as observed in individual rearing (refer to Appendix A for the actual date of death).

This study was conducted in an ambient environment, in which the temperature was kept mostly at 23–26 °C, but dropped to around 20 °C in winter. We observed that the egg and instar periods tended to increase below 24 °C and at/below 22 °C, respectively. It is clear that these tendencies are due to temperature changes. Such delay in developmental time is reported in other Lepismatids. In *C. longicaudatum*, which prefers temperatures between 20 and 26 °C, development slows down largely below 20 °C [15,16]. *T. domestica* has a higher optimum temperature, near 37 °C, and the average time of development from hatching to maturation was 92 days at 37 °C, and increased to 247 days at 29 °C [17]. In *L. saccharinum*, occurrence of diapause instars (lasting more than 50 days) was affected by temperature; at 28 °C, diapause was prevented, and, with lower temperatures, 25 °C and 20 °C, the frequency of diapause incidence increased [13]. Occurrence of diapause was also affected by photoperiod and age [13].

Although this study was not conducted under constant temperature and humidity, the temperature range in this study was deemed similar to those in exhibition halls and storage rooms in many museums in Japan. On the other hand, the RH range in this study, 60–70%, was slightly higher than in many museums in Japan, which normally control RH at 50–60%. However, we chose this RH range because we assumed that insects in museums would prefer higher RH and find small spaces, such as cracks, where the microclimate is more humid than in general display or storage spaces. Therefore, the obtained data should reflect biological characteristics of *C. calvum* in a realistic environment. However, further research under more strictly controlled temperature and humidity is needed to clarify the effects of environmental conditions on the developmental periods of eggs and each instar.

The growth ratios of HW were around 1.1 (Table 3). In detail, the growth ratios tended to increase until the 6th instar, and to decrease afterwards. A similar analysis based on HW between compound eyes was reported in a Japanese literature on *Ctenolepisma* sp. [3], and its growth ratios were mostly equivalent to those in *C. calvum* found in this study. It should be noted that this species is mentioned in some Japanese literature [2,3], and it is regarded as a different species from *C. villosum*, *C. lineatum*, and *C. longicaudatum* and is reported to be morphologically similar to *C. pinicola* Uchida (1964) (which has been synonymized with *C. longicaudatum* [18]) and *C. ciliatum* (Dufour, 1831) [3], but the specific name of this species is not given in these reports. The growth ratios in *C. calvum* were smaller than those in typical hemimetabolous and holometabolous insects (reported median values being 1.27 and 1.52, respectively [19]). Because the body length varied in certain instars and overlapped substantially among different instars (Table 3), we considered that it was preferable to use HW as a criterion of body size growth.

### 4.2. Feeding Habits

Here, although not being the main topic of analysis in this study, we discuss the feeding habits of *C. calvum* based on mass-culture and individual rearing. The bred individuals often fed on facial tissue, creating many amorphous holes. They also consumed cricket powder, soy powder, and dead individuals of *C. calvum*, suggesting that they can utilize proteins as nutrients, as in *L. saccharinum*, *T. domestica*, and *C. longicaudatum* [16,17,20]. Fecal pellets collected from mass-culture cages were also given to immature juveniles in individual rearing, based on previous reports suggesting that microbial symbionts are spread through feces in *T. domestica* [11,12], but we were unable to determine if they consumed the fecal pellets. After ecdysis events, exuviae were sometimes consumed and sometimes not. Fish tablets, which were used for breeding of *C. longicaudatum* [21], were also suitable for *C. calvum*. In addition, we observed that *C. calvum* consumed fibrous dust collected from the floor of a museum gallery, when given. Although not measured in this study, it is expected that consumption rates decrease at low temperatures, as seen in *L. saccharinum* [20], e.g., at/below 22 °C. Proportional consumption of dietary components may also change with temperature, as in this species [20].

### 4.3. Oviposition and Reproduction

The results showed that the ovipositor first appeared in females at the 8th or 9th instar, and females reached the reproductive stage at the 10th or 11th instar within one year after hatching (Appendix A). Oviposition by the females in the 10th–12th instars occurred twice at maximum, and the largest number of eggs laid at one time was 16. Because, throughout 2022, 78.2 eggs were laid per female born in 2020 or before (approximately two years old or older) on average, it was suggested that females older than 12th instar can lay more than 16 eggs at one time and/or can reproduce more than twice a year.

Compared to other Lepismatid species, sexual maturation in *C. calvum* occurred in about the same instars as in *L. saccharinum*, and slightly earlier than in *T. domestica* and *C. longicaudatum*. *L. saccharinum* started oviposition in the 9th–10th instars [13]. *C. longicaudatum* matures between 8th and 13th instars, and becomes adult from the 14th instar [15,16]. In *T. domestica*, about 10 instars were passed before the ovipositor was visible from the dorsal view, and first oviposition occurred at the 14th instar at earliest [14].

The results suggest that the strains of *C. calvum* found in Fukuoka and Tokyo, Japan, can reproduce parthenogenetically. It is still possible that male individuals are present in these areas at a low ratio. Parthenogenesis may also occur in populations in other areas of Japan, but this is to be investigated in the future. Parthenogenesis in Zygentoma was reported in *Nicoletia meinerti* Silvestri, 1905 (Family Nicoletiidae) [22], but, to the best of our knowledge, there are no reports on parthenogenesis in Lepismatidae. In Czechia, both sexes of *C. calvum* were found [9], and sex ratio bias is not mentioned in this report. Detailed observations of reproduction through successive generations and genetic analyses are necessary to understand the patterns and mechanisms of reproduction in *C. calvum* in Japan.

Lindsay [15] reported that the average number of eggs laid per year by a *C. longicaudatum* female was 56, fewer than a two-year old or older female of *C. calvum* observed in this study. Together with the possibility of parthenogenesis, *C. calvum* may have higher fecundity and potential for rapid spread than *C. longicaudatum* in Japan.

There was a clear seasonal trend in oviposition; it occurred from spring to autumn and the peak in oviposition was in early June (Figure 6). Temperature is a factor affecting the time of oviposition, as in *T. domestica*, in which oviposition occurred at 32–41 °C but did not occur at 27 °C or below [17]. However, the average daily temperatures that were stable from June to September (Figure 3) do not explain the decrease in the number of eggs laid in the latter half of this period, and, thus, there may be other factors that influence the seasonal trends of oviposition.

### 4.4. Possible Control Measures in Museums

Lowered temperature should slow down the development and, probably, feeding activity of *C. calvum*. Although it may not be easily adopted in many museums, keeping the temperature around 20 °C or below is a possible way for managing this species. As generally known, deep cleaning in the exhibition and storage rooms is important to remove sources of nutrients, such as fibrous organic materials and insects remains. In this study, it was shown that a large number of eggs were laid in early summer. They should hatch into 1st instar juveniles in approximately two months and grow into active juveniles by autumn. Therefore, it may be effective to apply an insecticide as a control measure, especially in summer and autumn, to prevent growth in the population size. An example of application of a pyrethroid emulsion by injection and with a brush to boundaries between flooring and walls, combined with deep cleaning, is demonstrated [23]. Dietary requirements and feeding preferences, after detailed investigations, can be used to develop effective poison bait. We have recently confirmed the insecticidal effect of a bait containing bistrifluron on *C. calvum* [24]. Validation of the efficacies of these suggested measures and development of poison baits for practical use are required.

## 5. Conclusions

This study provides new data on the egg-to-adult development and reproduction of the *C. calvum* strain found in Japan. The life cycle of the strain in an indoor, ambient environment can be summarized as follows. Oviposition occurs from April to November, with a peak in early June. Eggs hatch within two months, except in winter, when hatching takes longer. The juveniles grow though molting and mature sexually by the 10th or 11th instar within one year. Once matured, females can oviposit once or twice (or possibly more) a year, laying around 10 or more eggs at one time. It is suggested that *C. calvum* can reproduce parthenogenetically, at least in particular areas of Japan. Further research on their biology, such as responses to different environmental conditions, feeding habits, and patterns and mechanisms of reproduction, as well as methods for practical control, are necessary.

## Figures and Tables

**Figure 1 insects-14-00563-f001:**
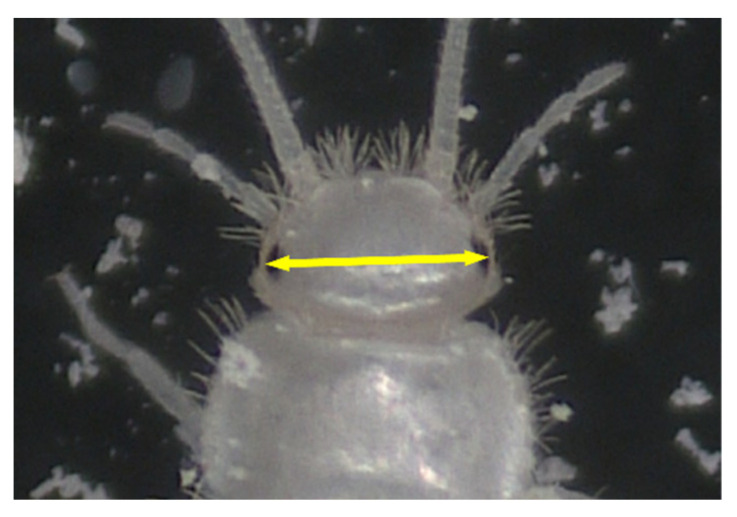
Distance (arrow) measured as the head width (HW) of *C. calvum*.

**Figure 2 insects-14-00563-f002:**
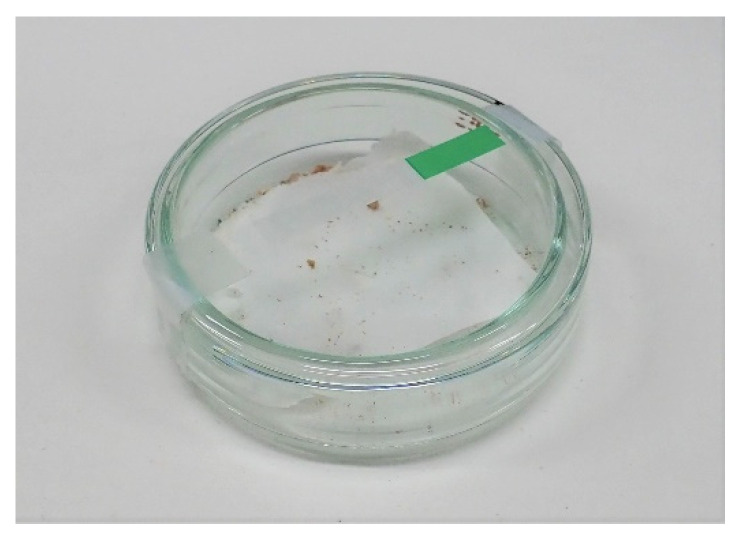
Petri dish used for individual rearing.

**Figure 3 insects-14-00563-f003:**
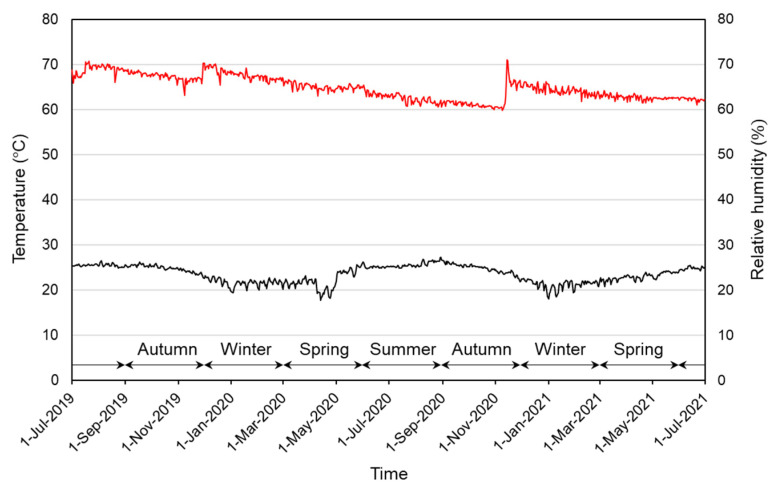
Average daily temperature (lower graph) and relative humidity (upper graph) in the plastic box where mass-culture cages and petri dishes were kept.

**Figure 4 insects-14-00563-f004:**
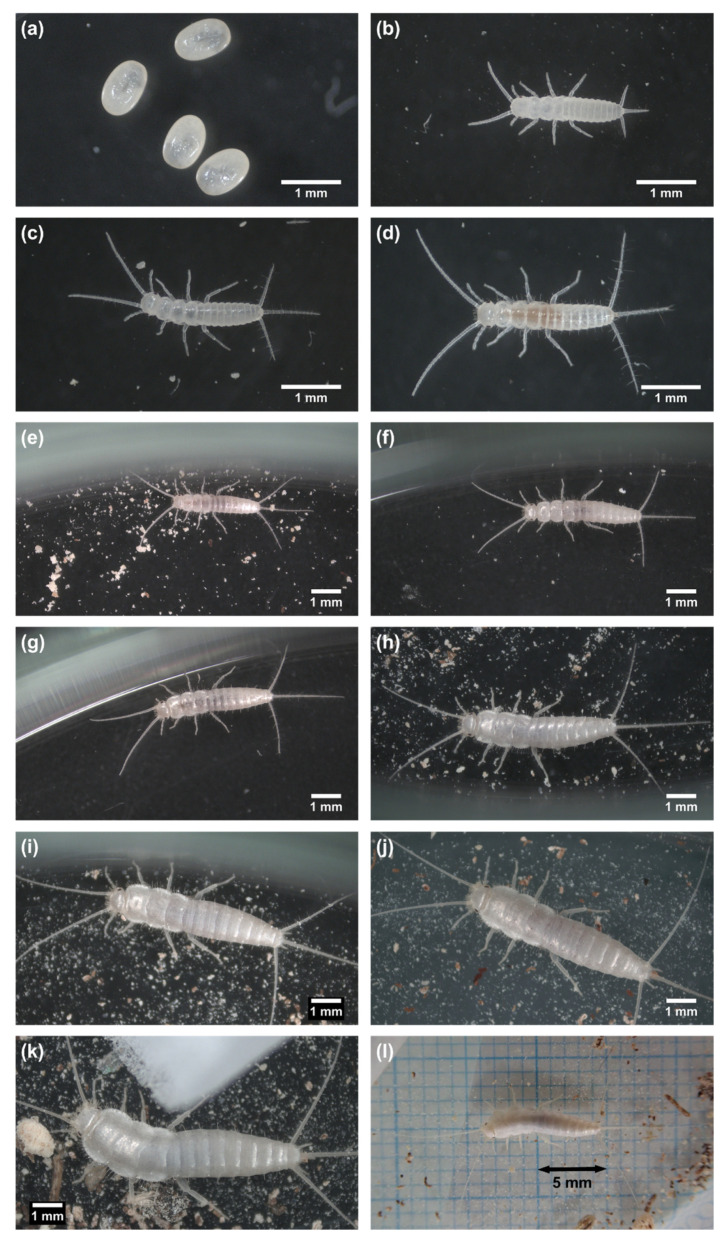
Developmental stages of *C. calvum*. (**a**) eggs; (**b**) 1st instar; (**c**) 2nd instar; (**d**) 3rd instar; (**e**) 4th instar; (**f**) 5th instar; (**g**) 6th instar; (**h**) 7th instar; (**i**) 8th instar; (**j**) 10th instar; (**k**) 13th instar; (**l**) entire body and appendages of the same individual as in (**k**). Note that the images in (**a**–**k**) are not necessarily of the same individual.

**Figure 5 insects-14-00563-f005:**
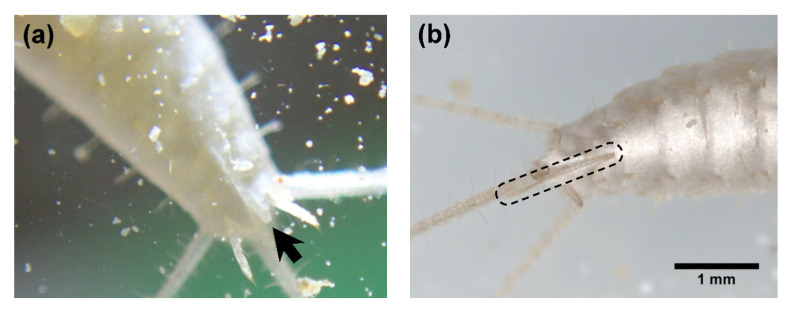
Photographs of the ovipositor. (**a**) Short ovipositor first observed at the 9th instar. Note that scale bar is not available in this photograph, but the ovipositor is estimated to be approximately 0.7 mm long; (**b**) fully developed ovipositor of a different individual (instar unknown).

**Figure 6 insects-14-00563-f006:**
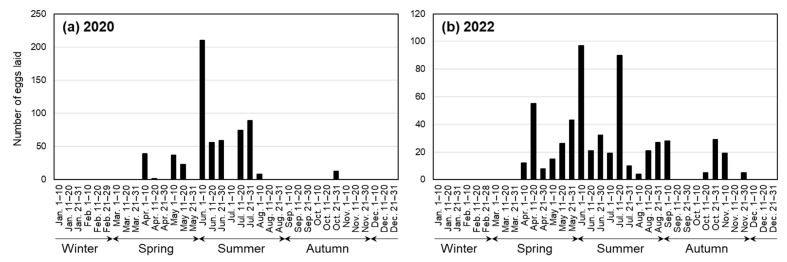
Seasonal trend in oviposition in a year. (**a**) Eggs collected in 2020 from mass-culture cages and petri dishes for individual rearing; (**b**) Eggs collected in 2022 from two cages containing a total of 17 females.

**Table 1 insects-14-00563-t001:** Average egg periods and survival rates of eggs and young (1st–2nd instar) juveniles.

Egg Duration (Mean ± se) (Days)		Survival Rate	
Avg. Temp > 24.0 °C	Avg. Temp. < 24.0 °C	Egg	1st–2nd Instar Juvenile
56.9 ± 0.3 (*n* = 30)	72.4 ± 1.5 (*n* = 36)	0.96 (*n* = 70)	0.78 (*n* = 40)

se standard error.

**Table 2 insects-14-00563-t002:** Instar durations (days) of each individual, with notes related to oviposition.

Individual Number	Instar Number														
	1	2	3	4	5	6	7	8	9	10	11	12	13	14	15
#1	n/m	n/m	n/m	29	17	19	21	33 *	*119*	*68*	37 **	d			
#2	n/m	n/m	n/m	28	17	20	25	35 *	*114*	*70*	35 **	d			
#3	n/m	n/m	n/m	39	26	20	23	39	*80* *	*76*	40 **	44	76	*151*	d
#4	n/m	n/m	n/m	65	20	20	29	*90*	*55* *	31	47 **	71	d		
#5	n/m	n/m	n/m	22	26	26	*46*	*60*	*70* *	d					
#6	n/m	n/m	n/m	21	22	28	*48*	*57*	*62* *	d					
#7	n/m	n/m	34	25	20	32	*53*	*46*	*55* *	** d					
#8	5	12	21	17	33	d									
#9	5	12	21	17	39	*81*	*69*	29 *	d						
#10	5	12	21	17	39	*105*	*48*	26 *	d						
#11	n/m	n/m	*44*	*42*	19	21	27	25	30 *	d					
#12	n/m	n/m	*44*	*42*	22	21	28	d							
#13	n/m	n/m	*44*	*42*	23	22	d								
#14	n/m	n/m	*44*	*42*	17	22	27	37	56 *	d					
#15	n/m	n/m	*44*	*42*	22	21	28	36	* d						
#16	n/m	n/m	*44*	*42*	22	21	28	* d							
#17	5	12	24	24	23	a/d									
#18	5	12	a/d												
#19	5	14	26	21	17	25	30	* d							
#20	5	14	27	20	22	21	32	ukn	ukn *	d					
#21	ukn	ukn	26	20	22	26	33	*87*	* d						
#22	4	14	27	19	22	20	28	* d							
#23	4	14	27	19	22	d									
#24	ukn	ukn	d												
#25	ukn	ukn	d												
#26–40: additional individuals for the observation of juvenile development until the 2nd–4th instars (see Appendix A for details)															

Italicized numbers indicate that the average temperature was 22.0 °C or below. n/m not measured (not under observation); ukn unknown, albeit under observation; d death that was considered natural; a/d accidental death by mishandling (not counted as natural death). * first observation of presence of ovipositor. ** first observation of oviposition.

**Table 3 insects-14-00563-t003:** Instar durations (mean ± se), head width (HW) (mean ± se) and growth ratio, approximate range of body length in each instar, and instar-specific survival rates at the 3rd and later instars.

Instar Number	1	2	3	4	5	6	7	8	9	10	11	12	13	14	15
Instar duration (days) at avg. temp. > 22.0 °C	4.7 ± 0.1	13.2 ± 0.4	26.1 ± 0.9	25.2 ± 3.0	23.3 ± 1.3	22.6 ± 0.9	27.6 ± 0.9	33 ± 2	43	31	40 ± 3	58	76	-	-
Instar duration (days) at avg. temp. ≤ 22.0 °C	-	-	44 ± 0	42 ± 0	-	93	53 ± 4	68 ± 9	79 ± 10	71 ± 2	-	-	-	151	-
HW (mm)	0.360 ± 0.002	0.398 ± 0.002	0.442 ± 0.002	0.495 ± 0.003	0.560 ± 0.005	0.641 ± 0.006	0.724 ± 0.008	0.803 ± 0.009	0.862 ± 0.010	0.929 ± 0.010	0.994 ± 0.011	1.057 ± 0.014	1.125	1.147	1.155
Growth ratio of HW	-	1.10	1.11	1.12	1.13	1.14	1.13	1.11	1.07	1.08	1.07	1.06	1.07 *	1.02 **	1.01 **
Approx. body length (mm)	1.7–2.0	1.8–2.3	1.9–3.0	2.2–3.4	2.9–4.2	2.9–4.9	3.9–5.2	4.3–6.4	4.9–6.9	5.4–7.2	6.3–7.2	6.8–7.8	7.7–8.6	approx. 8	approx. 8
Instar-specific survival rate	-	-	0.88	1.0	1.0	0.90	0.95	0.78	0.71	0.40	1	0.5	0.5	1	0

se standard error. * Ratio of average HW values of only two individuals, #3 and #4, that developed to the 13th instar. ** Ratio of HW values of only one individual, #3, that developed to the 14th and 15th instar.

**Table 4 insects-14-00563-t004:** Records of oviposition by individually reared females.

Individual Number	Date of Oviposition, Instar in Which Oviposition Occurred, and Number of Eggs Laid	
#1	12 June 2020, 11th instar, 16 eggs	
#2	15 June 2020, 11th instar, 8 eggs	21 July 2020, 12th instar, 7 eggs
#3	9–11 June 2020, 11th instar, 11 eggs	21–22 July 2020, 12th instar, 15 eggs
#4	19 June 2020, 11th instar, 11 eggs	
#7	11–16 June 2020, 10th instar, 6 eggs	

## Data Availability

Data is contained within the article and Appendix A.

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
