# Peer review of "Development and Reproduction of a Japanese Strain of Ctenolepisma calvum (Ritter, 1910) at Room Temperature"

_insects, 2023, doi:10.3390/insects14060563_

Round 1
Reviewer 1 Report
See attached file

Author Response
We wish to express our appreciation to the Reviewer for carefully reading our manuscript and giving fruitful suggestions. Our response has been uploaded as a separate file.

Reviewer 2 Report
Comments
1. The manuscript did not provide information about light as an environmental condition for rearing of this insect. Light conditions are also a factor that can affect the development of insects, so I think it is desirable to include information about them in this manuscript. The developmental and reproductive characteristics presented in this submission are pioneering findings for the insect species, but subsequent studies may yield more specific and differently-interpretable results for the same subjects. Therefore, in a general sense, it is recommended to describe the experimental conditions in more detail in order to create a reference point for comparison with the results of different studies in the future. In connection with this comment, please consider changing the manuscript title to the following: 'Development and reproduction of a Japanese strain of Ctenolepisma calvum (Ritter, 1910) (Zygentoma: Lepismatidae) under an unspecified condition'.
2. Although the development and reproductive characteristics of the experimented insect were described in detail, the manuscript did not present a comparison with the results of previous studies on this subject. Even if the research results on the characteristics of this insect species are being provided for the first time, it is necessary to present the characteristics of similar species together to prove that this manuscript results may not be deviated greatly from the natural characteristics of this insect. Please check information on biology of other species in the same genus, the same family or the same order, and include those ones into the discussion section through proper additional correction.
3. Please add the results about maximum life span and instar number, and average fecundity in the abstract if space permits.
4. The insect populations used in this experiment were collected from two regions in Japan. This manuscript did not provide information on the reproductive characteristics of populations collected from other regions. Therefore, I think it is not necessary to present the assumption that only parthenogenetic lines exist throughout Japan in the abstract. In this regard, consider replacing the sentence in line 30-31, page 1 with 'Through this study, only females were found, and the reproducible females produced their progenies parthenogenetically'. (For reference, it is difficult to conclude that thelytoky is occurring in the current laboratory colony because there is no evidence on whether all the individuals of the next generation developed to only females in the rearing experiment.) And, it is recommended that the related contents in the discussion section should be corrected.
Author Response

(The authors gave the same response as above.)

Reviewer 3 Report
Watanabe et al. presents data from an experiment assessing the effects of room temperature on the ghost silverfish development. Overall, the content is okay, and I think the paper is certainly publishable at some point. It seems to me that this is important, basic research on this highly detrimental indoor pest. This journal is perhaps appropriate, but I suggest resubmitting the work once the following corrections are made.
The Intro and Discussion provide no insight on how this MS relates to the various other ones cited in the text or concerns that have been raised by other researchers in the field. The authors do not present any hypotheses or expectations that could be connected to previous studies; adding these details will improve the paper. This article should provide details on all these fronts to provide the proper context for the work.
The authors should also give more details about the experiment and analyses in the Methods section. For example, the sampling methods seem sound, but data analysis and quantification of duration days and survival rates is not well explained. The results section is mainly descriptive, and several parts are missing or show inconsistencies. The resulting figures and tables are insufficient and of poor quality not helping to follow the reasoning throughout the manuscript. Oddly, no SEMs or error bars were shown or interpreted at all, and no sequential analyses were performed.
The discussion lacks real concluding remarks in my opinion, and if I was an IPM practitioner or consultant, I’d want to see these recommendations for my area. Adding this information would benefit the discussion.
Overall, I was excited to see the results of the paper after reading the abstract, but I found it hard to extract key messages useful to policymakers and professionals, probably in large part due to the lack of connection with other published work and need for improved structure of the current manuscript with respect to a wireworm species in place.
This is not to diminish the data gathered in this study, they are of value. The paper would benefit from a more thorough literature review and a better connection with more relevant reports on the subject.
The next draft of this paper will need to be dramatically different to have a chance at publication in my humble opinion.
Author Response

(The authors gave the same response as above.)

Round 2
Reviewer 3 Report
Some minor improvements have been made by authors to the paper following my original suggestions, and the study is interesting, nevertheless …
No changes were made to the introduction even though more connection to previous studies was suggested. The introduction and discussion provide no insight on how this MS relates to the various other ones cited in the text or concerns that have been raised by other works. Everything is there, but the authors should clearly explain in their introduction why the research was done, why it was important, and how it fits with these other studies. It should be clear and concise, and sadly it is not.
The statistics section is largely missing to support authors' claims, which additionally affected my ability to understand whether the authors have entirely interpreted their results, and the study is primarily descriptive. The resulting tables are still insufficient or inconsistent (in some cases averages where presented and in other cases ranges were reported, with or without the SEMs, etc., etc.) and could be improved to help follow the reasoning throughout the manuscript. Photos are actually quite impressive and of great quality!
The discussion should explain the significance of the results and place them into a broader context, and it is not. It should not be redundant with the results section, and it is. Redundancy made reading this paper tedious and lengthy.
I will recommend this paper to be published as a technical report or short communication after revising with what I had in mind here, but not as a separate research article.